# Longitudinal Structure–Function Evaluation in a Patient with *CDHR1*-Associated Retinal Dystrophy: Progressive Visual Function Loss with Retinal Remodeling

**DOI:** 10.3390/diagnostics13030392

**Published:** 2023-01-20

**Authors:** Andrea Cusumano, Benedetto Falsini, Fabian D’Apolito, Michele D’Ambrosio, Jacopo Sebastiani, Raffaella Cascella, Shila Barati, Emiliano Giardina

**Affiliations:** 1Department of Ophthalmology, Tor Vergata University, 00133 Rome, Italy; 2Macula & Genoma Foundation, 00133 Rome, Italy; 3Department of Ophthalmology, Policlinico A. Gemelli, IRCCS/Catholic University, 00133 Rome, Italy; 4Department of Biomedicine and Prevention, Tor Vergata University, 00133 Rome, Italy; 5Genomic Medicine Laboratory-UILDM, Santa Lucia Foundation IRCCS, 00179 Rome, Italy; 6Department of Biomedical Sciences, Catholic University Our Lady of Good Counsel, 1000 Tirana, Albania

**Keywords:** cone–rod dystrophy, *CDHR1* gene, visual function, retinal structure, inner retina, retinal remodeling

## Abstract

Background: Retinal dystrophies related to damaging variants in the cadherin-related family member 1 (*CDHR1*) gene are rare and phenotypically heterogeneous. Here, we report a longitudinal (three-year) structure–function evaluation of a patient with a *CDHR1*-related retinal dystrophy. Methods: A 14-year-old girl was evaluated between 2019 and 2022. An ophthalmological assessment, including color vision, perimetry, electroretinography, and multimodal imaging of the retina, was performed periodically every six months. Next-generation sequencing disclosed two likely pathogenic/pathogenic variants in the *CDHR1* gene, in compound heterozygosity, confirmed by segregation analysis. Results: At first examination, the patient showed a cone–rod pattern retinal dystrophy. Over follow-up, there was a decline of visual acuity and perimetric sensitivity (by ≥0.3 and 0.6 log units, respectively). Visual loss was associated with a progressive increase in inner retinal thickness (by 30%). Outer retina showed no detectable changes over the follow-up. Conclusions: The results indicate that, in this patient with a *CDHR1*-related cone–rod dystrophy, the progression to severe visual loss was paralleled by a progressive inner retinal thickening, likely a reflection of remodeling. Inner retinal changes over time may be functionally relevant in view of the therapeutic attempts based on gene therapy or stem cells to mitigate photoreceptor loss.

## 1. Introduction

The retinal dystrophies associated with damaging variants in the cadherin-related family member 1 (*CDHR1*) gene, located on 10q23.1 and coding for a photoreceptor-specific cadherin, are rare (it has been estimated by Stone et al. [1] that in the US the disease affects about 700 individuals, with an incidence rate of nine cases/year) and phenotypically heterogeneous. The *CDHR1* protein is a structural protein localized at the base of the rod and cone photoreceptors’ outer segments, and plays a fundamental role in the maintenance of cellular structural integrity [2].

Diverse clinical retinal phenotypes have been associated with *CDHR1* pathogenic variants, including cone–rod and rod–cone dystrophies as well as macular dystrophy and late-onset macular dystrophy [3,4,5]. Macular dystrophy is often present. While age of onset might be a prognostic factor for disease progression and severity, no clear effect of age on progression of disease has been reported [3,4,5]. In several patients, severe visual dysfunction (with visual acuity and color vision loss) has been associated with a relatively mild structural damage [5]. On the other hand, progressive thinning of the outer retina has been reported in a few patients over a variable period of follow-up [5]. The structure–function relationship, or lack of it, has not been unequivocally and clearly reported.

In the present study, we report a longitudinal structure–function evaluation of a young patient with *CDHR1*-related retinal dystrophy over a three-year interval. The results showed progressive visual function loss in presence of a retinal remodeling, as evidenced by thickening of the inner retina.

## 2. Methods

The proband was evaluated from 2019 to 2022, with annual visits. A complete ophthalmological evaluation, including refraction, anterior segment biomicroscopy, Goldmann applanation tonometry, and dilated slit lamp biomicroscopy, was performed at every visit. In addition, we performed color vision by Ishihara plates, computerized perimetry for central retinal function by Zeiss Humphrey frequency doubling technology (FDT) with mean deviation and pattern standard deviation reporting (Hymphrey software) as well as kinetic perimetry by Goldmann V/4e size, flash electroretinography according to a published technique for diagnosis, and follow-up of patients with cone–rod dystrophy [6], multimodal imaging by spectral domain OCT (Heidelberg), blue light and infrared autofluorescence, and multicolor fundus photography. We used a Heidelberg Spectralis HRA-OCT multimodal platform (Heidelberg Spectralis HRA-OCT 1.10.12.0) with 30° central lens for OCT images and near-infrared AF.

OCT assessment of inner nuclear layer (INL)/outer nuclear layer (ONL) thickness was manually achieved by two independent observers and disagreement was solved by consensus.

### 2.1. Genetic Analysis

Genomic DNA was extracted from buccal swab [7] using MagPurix Forensic DNA Extraction Kit and MagPurix Automatic Extraction System (Resnova) according to the manufacturer’s instructions. The concentration and quality of the extracted DNA was checked by DeNovix Spectrophotometer (Resnova). A panel-based next-generation sequencing (NGS) was used to screen candidate variants in IRD-associated genes.

The extracted DNA was sequenced using NextSeq 550 (Illumina), and the library preparation was performed on 20–50 ng/μL of DNA using Illumina DNA Prep with Enrichment and Tagmentation according to the manufacturer’s instructions. The obtained libraries were sequenced at 2 × 100 bp and the sequencing quality of the resulting data is expected to reach a quality score > 30 (Q30) for ~80% of total called bases. For the resulting variants, only those reporting a minimum coverage of 20X were considered eligible for the bioinformatic analysis. The sample was subjected to whole-exome sequencing (WES) and we successively created a virtual gene panel for the analysis of this phenotype. In particular, this virtual gene panel includes a number of 80 genes associated with inherited retinal diseases (IRDs). Moreover, the functional annotation of detected variants was performed by means of BaseSpace Variant Interpreter v. 2.15.0.110 (Illumina) and wANNOVAR tool (https://wannovar.wglab.org/, accessed on 1 June 2022). The interpretation of genetic variants was performed by publicly available reference databases (ClinVar, 1000 Genomes, GnomAD, Varsome). In particular, variants were classified and clinically interpreted according to the ACMG (American College of Medical Genetics and Genomics) guidelines [8], using the Varsome online platform (https://varsome.com, accessed on 1 June 2022).

The identified variant was confirmed by direct sequencing performed with BigDye Terminator v3.1, BigDyeX Terminator and ABI3130xl (ThermoFisher) according to the manufacturer’s instructions. In addition, direct sequencing was performed in order to conduct the segregation analysis in first-degree relatives.

This case study was performed according to the Declaration of Helsinki. The proband and her parents provided signed informed consent to anonymized data publication.

### 2.2. Case Report

The proband, a 14-year-old Caucasian female from Southern Italy, daughter of nonconsanguineous parents, was first examined in 2019. She had negative family history for retinal degeneration. She presented with visual acuity reduction in OU, not improving by correction of the myopic refractive error (-3 sph) found on retinoscopy (see Table 1, below, for details of the first examination). LogMAR visual acuity was 1.06 and 0.64 in OD and OS, respectively. No nystagmus was present. Color discrimination was absent by Ishihara plates in OU. Severe sensitivity loss of the central visual field was revealed by FDT in OU. OCT imaging showed abnormalities of the outer retina at the level of photoreceptor layer with unmeasurable ellipsoid zone in OU. Scotopic and photopic ERGs from both eyes were severely reduced in amplitude compared to normal control values, with more severe reduction for cone compared to rod mediated responses.

### 2.3. Genetic Results

The NGS analysis of the proband revealed the presence of three heterozygous variants. Two variants were localized in *CDHR1*, namely, NM_033100.4 (*CDHR1*): c.863-1G>A (rs886041900) and c.2012_2013del (p.Leu671Serfs*4) in intron 9 and 16 exon, respectively. The first is a single nucleotide variant with an MAF (minor allele frequency) of 3.99 × 10^−6^ in non-Finnish European population. The c.863-1G>A is absent in the gnomAD browser. The Variant Effect Predictor (VEP) classified the c.863-1G>A as splice acceptor variant which may disrupt canonical sites creating cryptic splice sites. Moreover, ClinVar, Varsome, and ACMG classification described this variant as pathogenic. Concerning c.2012_2013del, it is a novel nonsense variant (p.Leu671Serfs*4) which was predicted to be likely pathogenic (LP). This variant is not present in the literature or among online databases (ClinVar, gnomAD, Decipher). Prediction analysis described the c.2012_2013del as null variant, which may provoke nonsense-mediated decay (NMD), inducing the degradation of mRNA. The segregation analysis of the *CDHR1* gene showed that the mother was heterozygous for c.863-1G>A, while the father was heterozygous of c.2012_2013del.

Furthermore, the NGS analysis revealed the presence of a heterozygous variant in *ABCA4* gene (NM_000350: c.428C>T). The c.428C>T (p.Pro143Leu, rs62646860) is a single nucleotide variant with MAF of 7.95 × 10^−6^ in non-Finnish European population. The VEP described the c.428C>T as a missense variant with a damaging effect. In fact, ClinVar, Varsome, and ACMG classification described this variant as pathogenic. In addition, the *ABCA4* gene was also analyzed by multiplex ligation-dependent probe amplification (MLPA, detect copy number alterations), providing a negative result.

### 2.4. Clinical Results

Baseline and follow-up data of visual acuity, central visual field, OCT, and ERGs recorded from the patient are reported in Table 1. It can be noted that visual acuity declined progressively during the observation period, corresponding approximately to a ≥0.3 log MAR decline in OD and OS. Central visual field sensitivity also declined significantly in OU, by approximately 0.6 log units. The rate of decline was more severe for OD than for OS. Examples of FDT visual fields recorded from the left eye in 2019 and 2021 are shown in Figure 1. ERGs were severely reduced from baseline and remained reduced during follow-up, with a low signal-to-noise ratio. Examples of mixed rod–cone and cone ERGs recorded at baseline in 2019 and in 2022 are shown in Figure 2. OCT-measured central retinal thickness and volume tended to increase during follow-up.

Central retinal structure is shown in greater detail in Figure 3, Figure 4 and Figure 5. The linear and volumetric changes in outer retinal (Figure 3) and inner retinal thickness (Figure 4), obtained by automated segmentation, are reported for OD and OS, respectively. Figure 5 shows images of the inner retina at higher magnification, comparing a normal control eye with the *CDHR1* eye, recorded at baseline (2019), at the first (2020), the second, and third follow-up (2022). Note in the *CDHR1* eye the loss of normal lamination, specifically the reduced visibility of hyporeflective layers such as ONL and INL, and the progressive increase in the thickness of the inner retina.

Figure 6 shows plots of the trends of the manually measured average thickness of outer nuclear layer (ONL) and inner nuclear layer (INL), respectively, as a function of the age of the patient. The total retinal thickness values are also shown. The tables on the left report the individual measurements. It can be noted that INL thickness tended to increase over the follow-up period. The increase from baseline was by 20% in OD and 30% in OS. ONL thickness, by contrast, did not show substantial changes over the follow-up.

## 3. Discussion

In this report, we describe the longitudinal structure-function data of a young girl affected by cone-rod dystrophy associated with likely pathogenic and pathogenic variants in the *CDHR1* gene. This gene has been associated with autosomal recessive retinal dystrophy and plays a crucial role in maintaining photoreceptors structure and survival. The NGS analysis revealed that the proband resulted to be compound heterozygote for two variants (c.863-1G>A and c.2012_2013del) located on the *CDHR1* gene. In particular, c.863-1G>A and c.2012_2013del variants could have an effect on protein features as they provoke the creation of a cryptic splice sites and truncated protein (might cause NMD), respectively. In fact, these variants could trigger modifications in the structure and function of cadherin-related family member 1. In addition, the NGS analysis showed a pathogenic variant in the *ABCA4* gene at the heterozygous state. To this purpose, it is important to remember that the presence of a single variant in *ABCA4* gene is not consistent with this pathological phenotype, and, thus, this patient is also a carrier for *ABCA4* retinopathy, relevant in family planning.

The major finding of this study is that the visual loss, as measured by visual acuity and central visual field, was severe and progressive, and associated with progressive thinning of the outer retina, as evidenced by EZ, ELM loss, and ONL attenuation. As a secondary, nevertheless important, phenomenon, there was a measurable progressive increase in INL and inner retinal thickness.

Previous studies [5] described in a few *CDHR1* patients the follow-up results of retinal structure and function. These studies reported that, in some patients, progressive visual loss was not matched by progressive structural thinning of the retina, while other patients did show progressive retinal thinning. In the young patient described in this study, we found an opposite trend characterized by a compensatory/secondary thickening of the inner retina with outer retinal atrophic changes. We may speculate that the thickness increase during follow-up may reflect a remodeling of the retina, at inner structural level, as a consequence of early photoreceptor degeneration. This is a well-described phenomenon observed in several animal models of retinal degeneration [9] as well as in humans with different mutations causing retinal degeneration [9,10,11,12,13,14]. Jacobson et al. [11], evaluating inner retina in choroideremia patients, proposed that an early stage of remodeling can be characterized by an increase in the thickness of the inner retina. In our young patient, a progressive increase of inner retina thickness over a three-year interval was observed, consistent with the proposed model [11]. To our knowledge, this is the first report showing a progressive thickness increase, likely a feature of retinal remodeling, in human *CDHR1*-associated retinopathy.

The interest of the present case primarily lies in the young age and in the severity of the specific gene mutations. The product of the *CDHR1* is essential for photoreceptor structural stability and survival [2]. The protein loss may have induced an early-onset retinopathy involving primarily macular cone photoreceptors, as usually found in *CDHR1* patients, but also involving rods and peripheral cones, as shown by the profound flash ERG abnormalities. The photoreceptor dysfunction/loss resulted in severe visual function abnormalities (visual acuity, color vision, and perimetric sensitivity) paralleled by a progressive attenuation of outer retina and thickening of the inner retina. Inner retinal changes could reflect a localized inflammation causing swelling of inner retinal neurons. Alternatively, an abnormal upstream signal from photoreceptors may lead to autonomous electrical activity in the inner retina [15], an indirect sign of changes in synaptic connectivity and inner neurons. The scenario suggested by this specific correlation, or lack of correlation, between structure and function may be one where an abnormal “signal” from diseased photoreceptors triggers early and rapid changes in the inner retina [11,12]. These changes may precede inner retinal cell death with subsequent thinning that could be detectable in a longer follow-up. Inner retinal atrophy is likely a final common pathway of photoreceptor dysfunction in inherited retinal degeneration. Functionally, as reviewed by Telias et al. [15], the remodeling associated with outer retinal degeneration results in spontaneous intrinsic hyperactivity and membrane hyperpermeability in retinal ganglion cells. These changes might reduce visual functions to an amount possibly greater than that predicted by photoreceptor damage alone, and may be an obstacle to vision restoration in visually impaired individuals [15].

In conclusion, in this study we presented the results of a longitudinal structure/function evaluation in a patient affected by a relatively rare form of cone–rod dystrophy associated with likely pathogenic and pathogenic variants in the *CDHR1* gene. The data indicated that, in this patient, progression of early and severe functional loss was not paralleled by retinal thinning, as expected, but rather by retinal thickening, likely a feature of neuronal remodeling. Describing changes over time of structure and function may be of value not only at a diagnostic level, but also in view of the potential therapeutic attempts based on gene therapy or stem cells implants. These attempts should take into account the anatomic/functional changes occurring downstream to photoreceptors in the retina.

## Figures and Tables

**Figure 1 diagnostics-13-00392-f001:**
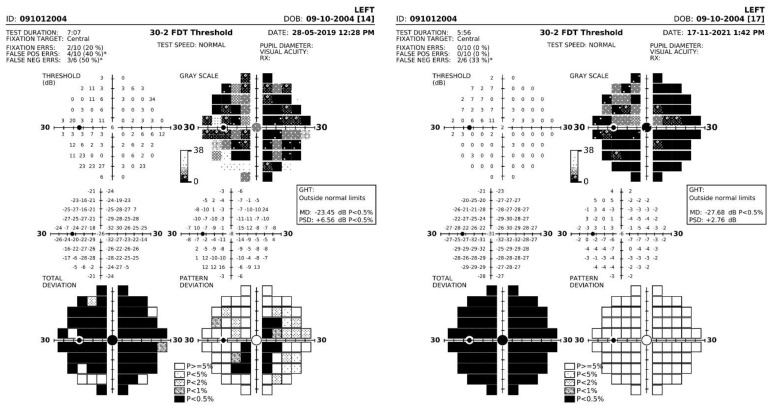
Examples of FDT visual fields recorded from the left eye in 2019 (**left**) and 2021 (**right**). Note that total deviation of sensitivity increased severely, as expressed by the changes in the grey scales of the total deviation plots (top row, **right**). The sensitivity loss was rather uniform across the visual field, as shown by the pattern deviation (bottom row, **right**), with no significant localized changes.

**Figure 2 diagnostics-13-00392-f002:**
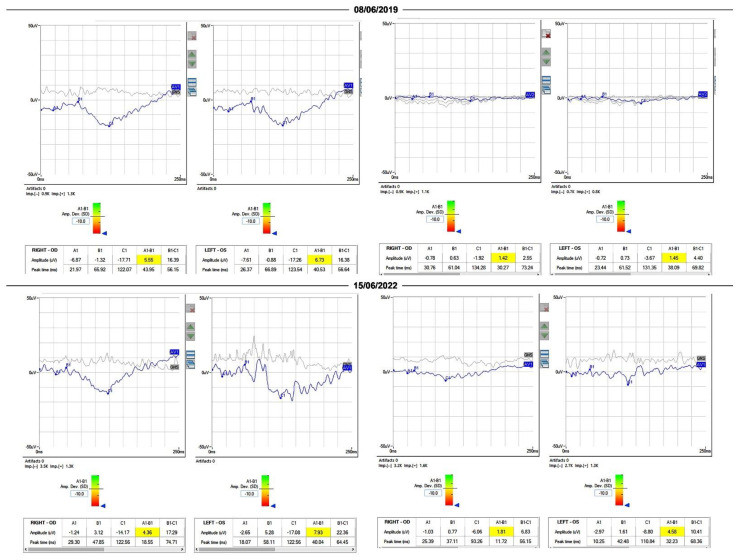
Examples of mixed rod–cone and cone ERGs recorded in 2019 and 2022. The records on the left in the figure are mixed rod–cone ERGs from OD and OS. The records on the right are cone ERGs from OD and OS. Mixed rod–cone and cone-mediated ERGs were of low amplitude and low signal-to-noise ratio during follow-up.

**Figure 3 diagnostics-13-00392-f003:**
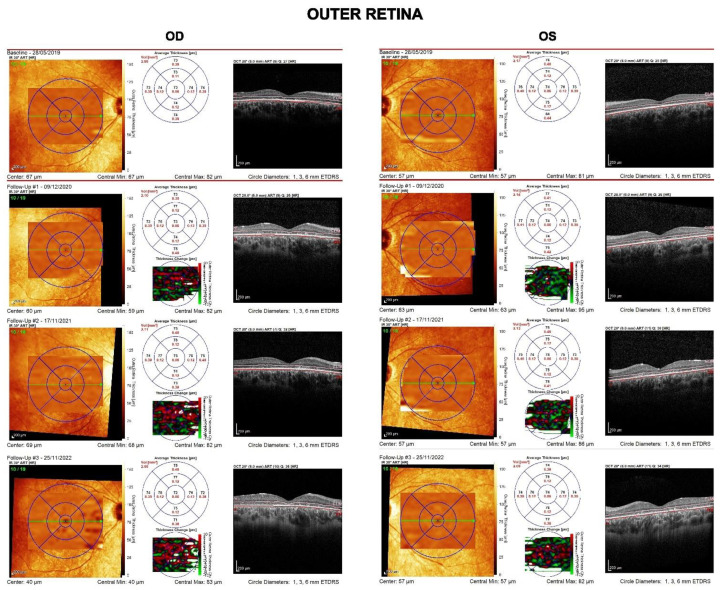
OCT imaging of the central retina (OD and OS) with plots of central outer retinal thickness and volume recorded in four separate visits over the follow-up period. Outer retinal thickness and volume were substantially unchanged during the follow-up.

**Figure 4 diagnostics-13-00392-f004:**
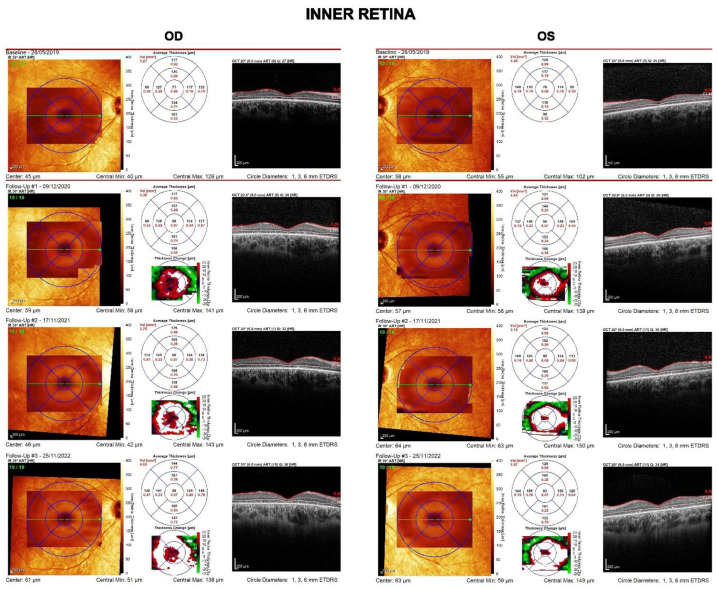
OCT imaging of the central retina (OD and OS) with plots of central inner retinal thickness and volume recorded in four separate visits over the follow-up period. Inner retinal thickness and volume tended to increase during the follow-up.

**Figure 5 diagnostics-13-00392-f005:**
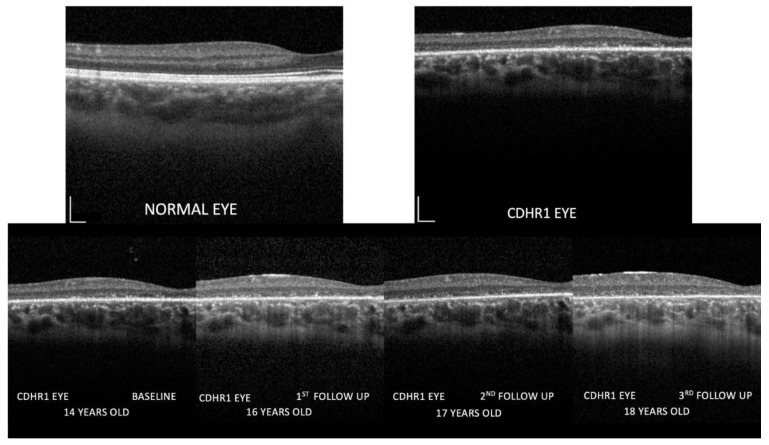
Images of the inner retina at higher magnification comparing a normal control eye with the CDHR1 eye (top row). Follow-up images recorded at baseline (2019), at the first (2020), the second, and the third follow-up (2022) are shown in the bottom row. Calibration bars indicate 200 micron both vertically and horizontally. Note in the *CDHR1* eye the progressively increasing outer retina abnormalities, the inner retina remodeling with loss of normal lamination, specifically the reduced visibility of hyporeflective layers such as ONL and INL, and the progressive increase in the thickness of the inner retina. No differences in ONL/INL thickness between nasal and temporal macula were observed.

**Figure 6 diagnostics-13-00392-f006:**
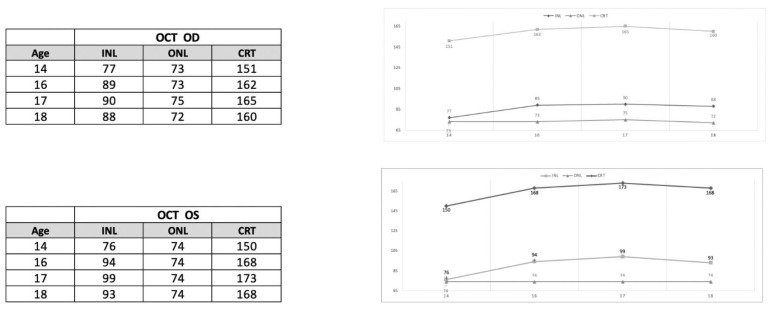
Plots of outer and inner retinal layer thickness as a function of age of the patient. Full retinal thickness is also shown for comparison. Data from OD and OS are reported. Note that while the mean outer nuclear layer thickness was substantially unchanged during follow-up in both eyes, inner nuclear layer thickness increased from baseline in OD and OS by 20 and 30%, respectively.

**Table 1 diagnostics-13-00392-t001:** Clinical data at baseline and during follow-up. Log MAR: logarithm of minimum angle of resolution; OCT = optical coherence tomography; CRT = central retinal thickness; VOLUME = central retinal volume; AF = blue field hypo-autofluorescence area in mm^2^; CONE = cone ERG amplitude; MIXED ROD-CONE = mixed rod–cone amplitude. CV FDT = visual field by frequency doubling technology. MD = mean deviation.

		BCVA OD	OCT OD			ERG OD				CV FDT OD
EXAM OD	AGE	LogMAR	CRT	VOLUME	AF (mm^2^)	CONE	PEAK TIME	MIXED ROD-CONE	PEAK TIME	MD
28/05/19	14	1.1	151	5.28						−19.43
08/06/19	14					1.77	38.09	5.09	65.92	
11/12/19	15	1.0								−27.46
01/09/20	15							8.31	66.21	
09/12/20	16	1.3	162	5.48	9.27					−31.32
11/01/21	16	1.4								
09/04/21	16	1.4								
15/06/21	16					4.04	61.33	9.16	64.45	
17/11/21	17	1.3	165	5.87						−31.24
15/06/22	17					1.81	37.11	4.36	47.85	
25/11/22	18	1.6	160	6.13	9.16					
		**BCVA OS**	**OCT OS**			**ERG OS**				**CV FDT OS**
**EXAM OS**	**AGE**	**LogMAR**	**CRT**	**VOLUME**	**AF (mm^2^)**	**CONE**	**PEAK TIME**	**MIXED ROD-CONE**	**PEAK TIME**	**MD**
28/05/19	14	53	150	5.4						−23.45
08/06/19	14					1.64	61.04	6.68	66.41	
11/12/19	15	55								−26.71
01/09/20	15					1.96	47.27	8.2	63.87	
09/12/20	16	22	168	5.63	7.45					−25.84
11/01/21	16	20								
09/04/21	16	20								
15/06/21	16					3.06	53.52	8.24	63.87	
17/11/21	17	20	173	5.82						−27.68
15/06/22	17					4.49	42.48	7.93	58.11	
25/11/22	18	20	168	5.96	8.01					

## Data Availability

Data available from the authors.

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
