# Peer review of "Longitudinal Structure–Function Evaluation in a Patient with *CDHR1*-Associated Retinal Dystrophy: Progressive Visual Function Loss with Retinal Remodeling"

_diagnostics, 2023, doi:10.3390/diagnostics13030392_

Round 1
Reviewer 1 Report
Please check the attachement

Author Response
Dear Editor
We would like to thank the Reviewers for their helpful and constructive comments on this paper. The changes we made and points we addressed are reported below:
Reviewer #1
Reviewer comments for Cusumano et al. Longitudinal Structure-Function Evaluation in a Patient with CDHR1-associated Retinal Dystrophy: Progressive Visual Function Loss with Retinal Re-modelling
The authors describe an interesting case of a young female affected by CDHR1-assocated AR retinopathy with clinical and genetic data. Their main finding is thickening of inner retina with less marked outer retinal changes. While the paper is interesting and the inner retina should be addressed with equal importance in IRD, I have concerns regarding their imaging findings which I have outlined below
Abstract:
Was she 15 at the first visit or the last. In table 1, she was 14 at first visit, please amend abstract age. done
How frequent were the visits? 6-monthly? Annually? 6 months (specified)
Suggest change wording of: ‘...inner retinal thickening, likely A REFLECTION of re-modelling.’ done
Change ‘vicar’ to ‘mitigate’ or similar. done
Introduction:
Very clear, well done. Thanks
Paragraph 2, last line, change ‘univocally’ to ‘unequivocally’ done
Methods:
Please expand on the clinical exam (e.g., dilated slit lamp biomicroscopy, Goldmann applanation tonometry, etc.) done
Also expand the multimodal imaging. Which cameras were used and which imaging modalities (e.g., blue light vs near infrared AF, multicolour fundus photography). Also, please describe the techniques used to report mean deviations etc for visual fields and how OCT assessment of INL/ONL thickness was achieved (i.e., automated or manual). We specified as follows: . In addition, color vision by Ishihara plates, computerized perimetry for central retinal function by Zeiss Humphrey frequency doubling technology (FDT) with mean deviation and pattern standard deviation reporting (Hymphrey software) as well as kinetic perimetry by Goldmann V/4e size, flash electroretinography according to a published technique for diagnosis and follow-up of patients with cone-rod dystrophy [6], multimodal imaging by spectral domain OCT (Heidelberg), blue light and infrared autofluorescence and multicolor fundus photography. We used Heidelberg Spectralis HRA-OCT multimodal platform (Heidelberg Spectralis HRA-OCT 1.10.12.0) with 30° central lens for OCT images and near infrared AF.
OCT assessment of inner nuclear layer (INL)/outer nuclear layer (ONL) thickness was manually achieved by two independent observers and disagreement was solved by consensus.
Sequencing details: you describe the technical details of DNA extraction, sequencing tech and bioinformatic variant calling thoroughly, but I do not see a description of the approach. I imagine you used a panel-based next generation sequencing approach to screen for
candidate variants in IRD-associated genes. Could you please add a sentence on the content of the panel? We added the following sentence:
The sample was subjected to Whole Exome Sequencing (WES) and successively we have created a virtual gene panel for the analysis of this phenotype. In particular, this virtual gene panel include a number of 80 genes associated with Inherited Retinal Diseases (IRDs). Moreover, the functional annotation of detected variants was performed by means of BaseSpace Variant Interpreter v. 2.15.0.110 (Illumina) and wANNOVAR tool (https://wannovar.wglab.org/).
2.2 Case Report:
Add ‘non-consanguineous’ to her history if this is the case, as we are discussing AR disease. done
Mention family history and whether she had nystagmus (relevant for age of onset of visual dysfunction). done.
VA: more helpful to describe in LogMAR equivalent. done.
Table 1: Add the word ‘amplitude’ to the ‘Cone’ and ‘Mixed Rod-Cone’ responses. Also, explain abbreviations in the table legend. done
2.3 Genetic Results
Paragraph 1, ‘The segregation analysis of THE CDHR1 gene showed that the mother WAS heterozygous for c.863-1G>A, while the father was heterozygous of c.2012_2013del. done’
Paragraph 2, spell out all acronyms (other than gene names), in this case MLPA. done.
2.4 Clinical Results
Paragraph 2, line 2: move the reference to (Figure 4) after the word thickness CDHR1 should be italicized throughout. done
Inner retinal changes. Rather than saying ‘fusion of the inner and outer nuclear layers’ I would suggest describing the OCT appearance. e.g., ‘loss of normal lamination, specifically reduced visibility of hyporeflective layers such as ONL and INL. We changed the text according to Reviewer’s suggestion’
In photoreceptor diseases, such as IRD, typically ONL becomes thinner as PRs atrophy, so what you may be seeing is absence/collapse of ONL with compensatory enlargement of inner retinal hyper-reflective
layers. As the incoming signal decreases with decreased PR function, atrophy of the nuclear layers (ONL corresponding to PRs and INL corresponding to bipolar cells) is expected. We agree with the Reviewer and changed the text accordingly.
Figure 5. My main concern here is the control eye image appears to be temporal retina and the top right CDHR1 eye is nasal/peripapillary retina. Please use the same type of image: that was a mistake, we amended it in the new figure figure 5.
On this note, was there a difference noted in ONL/INL thickness between nasal and temporal macula? No, this was specified in the text
Please add age in years to each of the images in bottom row. done
Figure 6. Were these automated or manual measurements of ONL/INL thickness? If automated, they may be prone to error as you can see the delineation of the INL is much less clear as the retinal degeneration progresses. They were manual, now specified in the Methods.
Add the units (microns). Done
I would also suggest adding total retinal thickness at each visit, as the % of total retinal thickness represented by ONL and INL would be another interesting metric. Done, now total retinal thickness reported in the plot of Figure 6
Discussion
Paragraph 1, last line: ‘not consistent with this pathological phenotype’ and thus this patient is also a carrier for ABCA4-retinopathy, relevant in family planning. Change done
Paragraph 2: Careful review of the OCT images is required before such a definitive statement can be made. The OCT images in figure 5 all show evidence of outer retinal layer loss including EZ, ELM, and significant attenuation/loss of ONL. Yes there is thickening of inner retinal layers but there is also loss of outer retina, as expected for the location and function of the CDHR1 protein. These inner retinal features are likely a secondary phenomenon. We agree with the Reviewer and changed the paragraph accordingly
Paragraph 3: suggest change in keeping with previous comment. Compensatory/secondary thickening of inner retina with outer retinal atrophic changes. Suggest change from ‘expression’ of retinal re-modelling to ‘feature’ or similar as the inner retinal changes cannot be proven as a direct expression of the genetic variant/mutation. We changed the paragraph according to Reviewer’s suggestion
Paragraph 4: Regarding inner retinal changes, do you think there is an element of localized inflammation causing this swelling. Does the abnormal or absent upstream signal from photoreceptors lead to autonomous electrical activity in inner retina (bipolar and ganglion
cells)? Is this swelling a precursor to inner retinal cell death and subsequent thinning of these layers as well (with longer follow up). This could be discussed as a final common pathway of PR/RPE dysfunction in IRD. We considered the points raised by the Reviewer and included these points in the paragraph.
Conclusion, last sentence: ‘anatomic/functional changes occurring downstream to photoreceptors changes done
We hope that changes we made are in the right direction to improve the quality of the manuscript
Best regards

Reviewer 2 Report
The authors have found an interesting correlation of inner retinal thickening with CDHR1 eye phenotype. Because there is need of extensive research with this gene and its association with retinal dystrophy, and the previous research on the topic has been somewhat inconclusive, especially regarding the outer retina and rod involvement, as such there is not enough consensus on the structure-function relationship, the manuscript brings important clinical results as a clear report by the authors. I would recommend the publication of the manuscript as is.
Author Response
The authors have found an interesting correlation of inner retinal thickening with CDHR1 eye phenotype. Because there is need of extensive research with this gene and its association with retinal dystrophy, and the previous research on the topic has been somewhat inconclusive, especially regarding the outer retina and rod involvement, as such there is not enough consensus on the structure-function relationship, the manuscript brings important clinical results as a clear report by the authors. I would recommend the publication of the manuscript as is.
Thank you for the very favorable comments
Round 2
Reviewer 1 Report
The authors have adequately addressed my comments. Nice paper. Best of luck!